# Challenges in the Diagnosis and Management of Non-Severe Hemophilia

**DOI:** 10.3390/jcm11123322

**Published:** 2022-06-09

**Authors:** Estera Boeriu, Teodora Smaranda Arghirescu, Margit Serban, Jenel Marian Patrascu, Eugen Boia, Cristian Jinca, Wolfgang Schramm, Adina Traila, Cristina Emilia Ursu

**Affiliations:** 1Department of Pediatrics, Division of Onco-Hematology, Victor Babes University of Medicine and Pharmacy Timisoara, 300041 Timisoara, Romania; esteraboeriu@yahoo.com (E.B.); sarghirescu@yahoo.com (T.S.A.); cristian_jinca@yahoo.com (C.J.); 2Onco-Hematology Research Unit, Romanian Academy of Medical Sciences, Children Emergency Hospital “Louis Turcanu” Timisoara, European Hemophilia Treatment Centre, 300011 Timisoara, Romania; mserban@spitalcopiitm.ro; 3Department of Orthopedics, Victor Babes University of Medicine and Pharmacy Timisoara, 300041 Timisoara, Romania; jenelmarianp@yahoo.com; 4Department of Pediatric Surgery, Victor Babes University of Medicine and Pharmacy Timisoara, 300041 Timisoara, Romania; boia.eugen@umft.ro; 5Ludwig-Maximilians University (LMU) Rudolf Marx Stiftung Munich, 80539 München, Germany; wolfgang.schramm@med.uni-muenchen.de; 6Medical Centre for Evaluation Therapy, Medical Education and Rehabilitation of Children and Young Adults, European Hemophilia Treatment Centre, 305100 Buzias, Romania; adinatraila@yahoo.com

**Keywords:** non-severe hemophilia, surgery, anti-FVIII antibodies, von Willebrand disease

## Abstract

(1) Background: Mild and moderate hemophilia, synonymous with non-severe hemophilia (NSH), are of constant interest for the clinicians. Bleeding occurs usually after trauma, injury, surgery, or inhibitor development, sometimes leading to a shift of the clinical phenotype from mild to severe, even with life-threatening and unexpected outcomes. (2) Methods: We performed a retrospective observational study conducted on 112 persons with congenital coagulopathies, 26 of them with NSH, admitted to our clinic in the period 2000 to 2022. For the diagnosis, we used laboratory studies (complete blood cell count, coagulation assays, biochemistry, thromboelastography, genetic tests) and imaging investigations (X-ray, ultrasound, CT, MRI). We selected four cases confronted with pitfalls of diagnosis and evolution in order to illustrate the sometimes provocative field of NSH. (3) Results: Confronted with challenging cases with under-, missed or delayed diagnosis and severe consequences, we aimed at presenting four such selected cases with mild or moderate hemophilia, real pitfalls in our clinical activity. (4) Conclusions: In the field of NSH, if not timely recognized, tending sometimes to remain ignored by caregivers and patients themselves, we can be confronted with challenging diagnostic situations and life-threatening bleeds.

## 1. Introduction

Non-severe hemophilia (NSH) is defined by a deficient clotting activity of FVIII or IX, ranging between 0.05 and 0.40 IU/mL and 0.01 and 0.05 IU/mL in the mild and moderate form, respectively. It is frequently highly dependent on the economic status of the reporting country, ranging from 37–42.6% in middle-high to 9–33.6% in low-income countries [1,2,3,4,5]. Even if the literature data are not always concordant, similarly influenced by the medical performances, there are obvious differences in clinical expression between NSH and severe hemophilia (SH): the first bleed is recorded at a median age of 6.5 (3.8–18.2) years in mild, of 4 (1.6–7.0) in moderate, and of 1.0 (0.5–2.0) years in severe hemophilia and the median age at diagnosis is 28.6 months for mild hemophilia, compared with 9.0 months for moderate and 5.8 months for severe hemophilia [6,7,8,9,10]. Obtaining a correct diagnosis may sometimes be a real problem, with NSH falling into the group of neglected diseases [8]. Underdiagnosis or delayed diagnosis, lack of medical and patient’s awareness, and misdiagnosis based on unusual non-suggestive presentation or outcome are prerequisites of a true hurdle in the management of this disease [11,12]. A timely, and correct diagnosis is of utmost importance, taking into account that even non-severe hemophilia, such as other coagulopathies, depending on their bleeding phenotype, could require a prophylactic approach [13,14,15,16]. This framework supports our present objective to provide some insights into the challenging NSH field, presenting some evocative cases from our experience in a country with long-time inadequate therapy, cases that represented real diagnostic pitfalls for our general practitioners, and even for our hematologists.

## 2. Materials and Methods

We performed a retrospective descriptive observational study conducted on 112 persons with congenital coagulopathies admitted to our clinic in the period 2000–2022. Inclusion criteria for NSH (the level of FVIII/IX ranging between 0.05 and 0.40 IU/mL and 0.01 and 0.05 IU/mL, in the mild and moderate form, respectively) were met by 26 patients. From the diagnostic point of view, there were used routine coagulation tests: activated Partial Thromboplastin Time (aPTT), Prothrombin Time (PT), Prothrombin Index (PI), FVIII, FIX, and vWF (von Willebrand Factor) activity, vWF antigen, using ACL Elite Pro coagulation analyzer through nephelometry method. Viscoelastic properties (rate, strength, stability) were analyzed by a thrombelastograph System 5000 (Haemoscope Corporation). Several parameters were measured: reaction time (s) (R), kinetics (s) (K), and maximum amplitude (mm) (MA). We also performed imaging investigations such as X-ray, ultrasound, CT (computed tomography), and MRI (magnetic resonance imaging). We focused our attention on the patients with late and misdiagnosis, or unexpected challenging outcomes encountered only in 4 cases, excluding 22 other cases without these provocative issues in the field of diagnosis and evolution. We considered them of illustrative and instructive value aiming at the avoidance of delayed diagnosis and inadequate treatment with the life-threatening outcome.

## 3. Results

The characteristics of our study group are registered in Table 1.

Patient 1 (P1). A 19-year-old male patient, without significant personal or familial medical history, was addressed to our clinic with the assumption of a retroperitoneal tumor and femoral vein thrombosis, diagnosis based on inferior right leg pain with partial functional impotence and the presence of a tumoral mass in the right thigh, affirmatively developed within one month after a forced hyperextension of the right lower limb; basin-pelvis X-ray revealed on the right medial femoral trochanter a non-homogeneous condensation area (~2.5/9 cm) with ossification in the soft tissues (~2.5/4 cm); ultrasound investigation confirmed the same aspect of tumoral mass in contact with the iliac crest and ischion, associated with femoral vein thrombosis. Considering the presence of acute vein thrombosis, anticoagulant therapy with unfractionated heparin (UFH) was started and the patient was transferred to our clinic. The CT scan and MRI showed the presence of a heterogeneous bulky tumor (15.2/7.2/4.1 cm) with calcifications, located in the right iliopsoas muscle (Figure A1), exercising a compressive effect on the iliac and femoral vessels, with partially obstructive thrombus. Laboratory tests revealed normal complete blood cell count (CBC), mild inflammatory reaction, prolonged aPTT (54.6 s), and elevated D-dimers (1506 ng/mL). Facing the results of the imagistic investigations, we changed our perception of persistent prolonged aPTT due to heparin therapy, and taking into consideration also other explanations, we performed additional coagulometric assays. These revealed a shortened prothrombin consumption time (27–34 s), prolonged reaction time in the thrombelastogram-TEG (8.4 min), and repeatedly low FIX activity (6.8–10.2%). Accepting the diagnosis of mild hemophilia B, under hemostatic control with factor IX concentrate replacement, surgery was performed with the excision of the tumoral mass. The histopathological aspect pleaded for post-traumatic osteogenous myositis. During the long-term follow-up, the patient was bleeding-free.

Patient 2 (P2). A 5-year-old boy, without any significant personal or familial medical history, was presented to the pediatric surgery department complaining of intense abdominal pain and paleness, affirmatively occurring post-traumatic, after a bike accident. The abdominal CT revealed a hepatic subcapsular hypodense area on the diaphragmatic surface (7.8/8.5/3.5 cm) in the left hepatic lobe (LHL), a hypodense area that extended in-band along the intersection line to the visceral surface of the liver, which was interpreted as grade 4 post-traumatic liver injury. Emergency surgery was performed; during the operation (evacuation of the hematoma and suture of the injured liver and its capsule) and two days afterwards, the patient presented excessive bleeding, associated with posthemorrhagic severe anemia, hemoglobin (Hgb) and hematocrit (Hct) dropping to 6.3 g/dL and 21.7%, respectively, requiring transfusion correction. Considering the unexplained severe bleeding during- and post-surgery a more comprehensive coagulation workup was performed: prothrombin time (PT) was within the normal range (11–13.1 s), aPTT (44.6–52.1 s), and reaction time (8.2 min) in TEG persistently prolonged, prothrombin consumption time (28–30 s) shortened, and FIX (3.3–4.4%) with low activity. The diagnosis of moderate hemophilia B was established and FIX replacement therapy was initiated, with good post-operative and long-term clinical outcomes.

Patient 3 (P3). A 55-year-old male patient, known since childhood as having type 2 von Willebrand disease (vWD), with very mild symptoms, with one or even less annualized bleeding rate (ABR), was treated on-demand with FVIII containing vWF. Recently, he decided on a presurgical hemostatic investigation for a programmed major invasive intervention. Astonishingly, vWF antigen and activity were within normal ranges, but the FVIII level was repeatedly between 4 and 5.5%. Extended quantitative and qualitative evaluations of vWF multimers were not performed; they could have given helpful information for the exclusion of vWD type 2 [17]. The genetic testing revealed a pathogenic hemizygous F8: c.6977 G>A variant, identified in exon 26 of the F8 gene (ref seq NM_000132.3). This substitution is a missense variant predicted to result in the substitution of arginine by glutamine (F8:p.Arg2326Gln). The variant has been described previously as Arg2307Gln, in patients with mild hemophilia A and is accepted as a pathogenic variant according to the MutaDATABASE criteria [18,19,20].

Patient 4 (P4). A 42-year-old male with known mild hemophilia A (FVIII: C = 13%), having a twin brother with the same form of the disease, recorded a relative low ABR and annualized joint bleeding rate (AJBR), ≤2, being treated on-demand with different brands of factor VIII, depending on their accessibility on our market; he developed high titer inhibitors (51.2 Bethesda Units) with a very low level of FVIII (0.4%), approximately 40 days after orthopedic surgery (arthroscopic meniscectomy for post-traumatic rupture of the internal meniscus of the right knee), performed under 4 days replacement with plasma-derived FVIII administered as continuous infusion. He experienced a dramatic evolution: large subcutaneous, retro- and intraperitoneal, intrapleural, and urogenital bleeding and a massive right iliopsoas muscle hematoma (Figure A2), with severe posthemorrhagic anemia (Hgb 6.7 g/dL and Hct 22%). Moreover, he developed major muscular bleeding in the right lower limb with compartment syndrome. The outcome progressively improved under treatment consisting of rFVIIa, corticosteroids, and transfusions of packed red blood cells. The genetic investigation identified on exon 11 of the F8 gene the causative mutation of mild hemophilia, a c.1648C > T substitution. This base change predicts the replacement of the native Arginine residue at codon 550 with a Cysteine (p.Arg550Cys), being potentially associated with a mild phenotype and risk for inhibitor development [18,19].

## 4. Discussion

NSH is a quite heterogeneous group of hemophilia, a possible subject of delayed, under-, or misdiagnosis, which, in a deceitful manner, can sometimes develop severe dangerous events with risky outcomes [9,21,22].

In patients without a family history of hemophilia, without joint or mucocutaneous visible bleedings, presenting only abdominal symptomatology, with challenging imagistic results, diagnosis recognition can be difficult and mostly exclusively based on laboratory exploration. Even in high-resource countries, the first bleeding suggestive of coagulopathy can occur as late as 18.2 and 7 years, in mild and moderate hemophilia, respectively [3,4]. In the Dynamo study, the median age of first joint bleeding was 7 years in moderate and 13 years in mild hemophilia [16]. Anesthesiologists in accordance with the American Society of Anesthesiologists and the British Committee for Standards in Hematology [23,24] play a critical role in the decision-making on perioperative exploratory protocols. Many of them consider that current evidence-based studies do not support routine unselected coagulation testing if the clinical history and physical examination do not predict a hemorrhagic event [23,24,25,26,27,28]. Therefore, screening tests, such as aPTT, PT, and platelet count, should be ordered only in evocative patients for bleeding risk [23,26,27,28,29]. In both patients (P 1, P 2) the clinical image was not suggestive of coagulopathy and the major invasive surgery without replacement therapy could have had a potentially disastrous outcome. Irrespective of the used reagents, aPTT is reasonably sensitive and specific for the situation of FVIII activity of 0.03 IU/mL [28]. In both cases, aPTT before starting replacement with coagulation factor had pathological values.

The occurrence of inhibitor development in mild hemophilia is low, with the global frequency ranging between 2.7 and 13% [30,31,32,33]. It gained more attention in the past decades, along with the more intensive genotyping in hemophilia [23,24,30,31]. The probability is much lower in previously treated patients (PTP), as in our P4; the patient had a negative family history of inhibitor occurrence, as his twin brother has also the same mild hemophilia without inhibitors. His missense mutation supporting the diagnosis of mild hemophilia is reported in the literature with risk of inhibitor development [18,19]. In our case, unexpectedly, after more than 150 ED, the benign bleeding pattern changed radically developing compartment syndrome and massive intra-abdominal and intra-pleural hemorrhage, associated with severe anemia, with a typical behavior of type 2 inhibitors. His genetic background could have contributed to this risky evolution. But considering the inhibitor negativity of the twin brother, there are probably the environmental factors of greater importance: the history of frequent switching of coagulation factor (CF) products used, because of low accessibility to them, endogenous danger signal represented by surgery, an increased dose of medication for surgery, and also possibly its administration as intravenous perfusion [32,33].

The importance of accurate examinations performed in an expert hemostasis laboratory, including vWF multimers exploration and chromogenic assays for the situation of therapy with EHL or non-factor treatments, is largely accepted [34,35,36,37]. Chromogenic assays, besides other advantages (lack of influence of lupus anticoagulant and of different clotting specific reagents), are useful for avoiding the important discrepancies between one-stage and chromogenic methods results that can lead to misdiagnosis of hemophilia phenotype. [36,38,39]. Biomolecular genetic analysis is also of utmost importance, as outlined also in P3. The absence of significant family history, with a very mild pattern of expression and the awareness of minor vWD, diagnosis established decades before, in times with a lack of adequate tests, justified a non-adherence to the monitoring program; the patient returned only recently prior to a programmed major surgical intervention. Inconsistencies of laboratory results require the exploration in a dedicated hemostasis laboratory with vWF multimers assay and the association of genetic investigations. In our case, without proper monitoring, the revealed missense biomolecular mutation, accepted as a pathogenic variant for mild hemophilia excluded the long-lasting prior assumed vWD.

We consider the small case series sample size and the lack of long-term follow-up of the selected patients as the limitations of our study.

## 5. Conclusions

Delayed-, under-, and misdiagnosis are not uncommon in the field of NSH, tending sometimes to remain neglected by caregivers and patients themselves. If not timely recognized, it can lead to severe, even life-threatening complications. The cases we presented wish to draw the attention to some clinical aspects: diagnosis, follow-up, treatment. The motivation is to plead for: improvement of diagnosis (more frequent routine indications of accurate coagulation investigation in challenging situations, introduction of vWF multimers analysis and chromogenic assay, increased usage of genotyping) for a better understanding of the pathogenic substrate and for identifying the likelihood of inhibitor development. Last but not least it is important the encouragement of persons with non-severe coagulopathies to respect the terms of follow-up, to assess earlier changes in diagnosis and therapy.

## Figures and Tables

**Table 1 jcm-11-03322-t001:** Characteristics of the case series.

Case	Initial Diagnosis	Definitive Diagnosis	Family History	Age/First Bleed (yrs)	Age/First Joint Bleed (yrs)	Age of Definitive Diagnosis/Complication (yrs)
P1,19 yrs male	retroperitoneal tumor, femoral vein thrombosis, and bleeding post heparin therapy	mild HB (6.8–10.7%), osteogenic myositis	no	19	no	19
P2, 5 yrs male	grade 4 traumatic liver injury with hepatic rupture	moderate HB (3.3–4.4%)	no	5	no	5
P3, 55 yrs male	vWD type 2	moderate HA (4–5.5%)	unconfirmed female cousin with mild vWD	4	21	55
P4, 42 yrs male	mild HA	high titer type 2 anti-fVIII inhibitors, (PTP, >150 ED), mild HA (11–13%), large subcutaneous, retro- and intraperitoneal, intrapleural, and urogenital bleeding, and a massive right iliopsoas muscle hematoma, and compartment syndrome	twin brother with mild HA	3	8	42 (high titer type 2 anti-fVIII inhibitors development)

HA—hemophilia A, HB—hemophilia B, vWD—von Willebrand Disease, PTP—previously treated patient, ED—exposure days, yrs—years.

## Data Availability

Not applicable.

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
