# Peer review of "Challenges in the Diagnosis and Management of Non-Severe Hemophilia"

_jcm, 2022, doi:10.3390/jcm11123322_

Round 1

Reviewer 1 Report

The manuscript focuses on moderate and mild hemophilia. Mild hemophilia is defined by factor levels between 5 and 40 IU/dL and is characterised by traumatic bleeds. Major issues associated with mild hemophilia are that it may not present for many years after birth, and that awareness, even within families, may be low.

Major commens:

- In the Discussion- The title of the manuscript focuses on management but lacks a description of prophylactic treatment. Higher factor levels (above 10 IU/dL) may be optimal to prevent subclinical bleeding. The prevention of bleeding is a key factor in progression of joint damage in patients with bleeding disorders. Recurrent joint bleeds, known as haemophilic arthropathy. This is also the case in other bleeding disorders, for example, a patient with congenital afibrinogenaemia, who experienced recurrent microbleeds into the joint, led to joint replacement in young age. Long-term prophylactic replacement significantly reduced bleeding versus on-demand treatment. The authors should cite two important manuscripts that described those facts:

- Simurda et al. Blood Coagul Fibrinolysis. 2015;26:978-80. doi: 10.1097/MBC.0000000000000392

- Miesbach W, et al. Thromb Res. 2021;199:67-74. 10.1016/j.thromres.2020.12.030.

- Page 4, line 181-194. The authors described von Willebrand's disease in this section, it is very important to state that it is in the diagnosis of VWD that the examination of vWF multimers is involved. HMW multimers and monitoring their quality and quantity help diagnose VWD and can be used to evaluate VWD treatment and to monitor response to treatment. Authors should cite the manuscript that described this: Skornova I, et al. Diagnostics (Basel). 2021 Nov 20;11(11):2153. doi: 10.3390/diagnostics11112153.

The methodological page is focused on a retrospective analysis of 112 people with coagulopathy. There are only 4 cases in the results.

Figures in the text are very clearly written. Authors should add the results of case reports to the table

I have to say that with these 30 references. Only 3 references are in the last 5 years. Authors should also add newer references

Author Response

Dear reviewer, 

We introduced the data regarding the prophylactic treatment and we added the references focused on prophylactic treatment issues. (Raw 50-52)

Also, we introduce the data regarding the importance of vWF multimers exploration, corresponding to raw 131-133, and the reference 17th, connected to this issue.

In the methodology, we mentioned that our study is a case series analysis with inclusion and exclusion criteria met only in 4 patients out of 26. NHS cases (raw24, 58-64).

As was recommended we prepared a short table with the patient’s characteristics (raw 78-81).

We added 7 references, as recommended, all published in the last 5 years.  (number 9, 10, 13-17)

Please address all correspondence concerning this manuscript at [email protected], [email protected].

We are grateful for your recommendations. Thank you for your consideration of this manuscript.

Sincerely,

Boeriu Estera, Ursu Cristina Emilia, and all authors of the manuscript

Reviewer 2 Report

The study is very interesting but it is necessary to improve conclusions because are very short without references that support it.

I recommend introducing a section with limits.

Author Response

Dear reviewer, 

In order to have an educational or instructive impact, we connected our conclusions to the presented cases with the conclusion.

Please address all correspondence concerning this manuscript at [email protected], [email protected].

We are grateful for your recommendations. Thank you for your consideration of this manuscript.

Sincerely,

Boeriu Estera, Ursu Cristina Emilia, and all authors of the manuscript

Reviewer 3 Report

While this is an interesting effort to highlight the challenges in non-severe hemophilia, it should not be presented as a research study. While the methods claim this is a retrospective observational study, it is actually a selective illustration of four cases. There is no criteria for inclusion or exclusion of cases from the initial database of 112, and no explanation for why 22 of 26 NSH cases were not described. It would be more appropriate to call this an illustrative case series.

The case presentations are not always clear, and it is difficult to determine the important take-away elements from each case. One case (3) is purely a lab related discussion while the others had concerning or confusing symptoms. The cases should be written to highlight the specific teaching/learning point.

The first paragraph of the discussion provides a good review of the challenges. The final sentence is not appropriate - every patient does not need a pre-surgical hemostatic assessment, but it should definitely be considered if there are bleeding symptoms or a personal/family history. The second paragraph (pertaining to case 4) is unclearly written and very speculative in tone; it would be best to stick to the evidence of inhibitor risk. The final paragraph does not tie into any of the cases; chromogenic assays may be valuable but are not essential if clot-based testing is used well.

The conclusion is broad, and the final point is not connected in any way to the cases, although it is good advice.

Author Response

Dear reviewer, 

We fully agree with you. Therefore, we mentioned that our study is a retrospective descriptive case series analysis. (raw 58-64)

We tried to improve our case presentations with the purpose to underline the real cause of mis- or late interpretations of the patient’s diagnosis and monitoring.

Regarding the pre-surgical haemostatic assessment, we deleted the last sentence, assuming your personal proposal to which we fully agree. (raw 180-181)

Regarding the inhibitor risk, we underlined the main contribution of environmental factors. (raw 193-198)

Please address all correspondence concerning this manuscript at [email protected], [email protected].

We are grateful for your recommendations. Thank you for your consideration of this manuscript.

Sincerely,

Boeriu Estera, Ursu Cristina Emilia, and all authors of the manuscript

Round 2

Reviewer 1 Report

The presented manuscript has been corrected in response to the suggestions. The authors have followed the recommendations of the reviewer. After the revision, the provided data and addition of the results became more clear. I would like to thank the authors for resubmitting the manuscript and explaining the obscure points from the previous version.

Author Response

Dear reviewer,

We tried to improve the method and we attached the table with corrections for more clear presentation of the cases, and we made changes to conclusions in order to be supported by the results.

Please address all correspondence concerning this manuscript to [email protected], [email protected].

We are grateful for your recommendations. Thank you for your consideration of this manuscript.

Sincerely,

Boeriu Estera, Ursu Cristina Emilia, and all authors of the manuscript

Reviewer 3 Report

I appreciate the editing that has been done to the paper, which has helped with some clarity. However, this is still not a case series but an illustrative selection of specific cases. There is an effort to improve the definition of the included cases, but the criteria of provocative issues is too broad as to be meaningful for a case series. It would be more appropriate to acknowledge that four cases seen in the center have been selected to illustrate key challenges in the recognition and diagnosis of NSH.

I suspect that the table was added to highlight the salient features of the four cases, but the information contained within it is not additive to the text, which has not been edited aside from the added sentence about vWF multimer testing. The table could have a column for the 'diagnostic challenge', or more simply each case could begin with a subheading to highlight the key issue.

The discussion and conclusion have been improved and better focused, but the final point is still not well connected to the cases. Patient advocacy for management (patient-centered care) is not relevant to the prevailing discussion of diagnostic uncertainty and poor recognition.

Author Response

Dear reviewer,

Changing the methods, and supplementing the results within the table, we have tried to comply with your requirements and improve the research design.

We accept your proposal, to make clearer “the provocative” issues characteristics for our 4 selected patients, recording under-, mis- and late-diagnosis, and unexpected challenging evolution.

Regarding the table, we added a column for the initial diagnosis in order to underline the challenge of establishing the definitive diagnosis.

We took into consideration your proposal regarding the final point of the conclusions and connected it with the lack of follow-up and the long duration of time (more than 10 years) without monitoring of P3.

Please address all correspondence concerning this manuscript to [email protected], [email protected].

We are grateful for your recommendations. Thank you for your consideration of this manuscript.

Sincerely,

Boeriu Estera, Ursu Cristina Emilia, and all authors of the manuscript